# Uromodulin and Vesico-Ureteral Reflux: A Genetic Study

**DOI:** 10.3390/biomedicines11020509

**Published:** 2023-02-10

**Authors:** Silvio Maringhini, Rosa Cusumano, Ciro Corrado, Giuseppe Puccio, Giovanni Pavone, Maria Michela D’Alessandro, Maria Chiara Sapia, Olivier Devuyst, Serena Abbate

**Affiliations:** 1Department of Pediatrics, Istituto Mediterraneo per i Trapianti e Terapie ad Alta Specializzazione ISMETT, 90127 Palermo, Italy; 2Pediatric Nephrology Unit, Azienda di Rilievo Nazionale ed Alta Specializzazione (ARNAS) Civico, Di Cristina, Benfratelli, 90129 Palermo, Italy; 3Department of Physiology, University of Zurich, CH-8057 Zurich, Switzerland

**Keywords:** uromodulin (UMOD), vesico-ureteral reflux (VUR), urinary tract infection (UTI)

## Abstract

Vesicoureteral reflux (VUR) is associated with urinary tract infections (UTI) and renal scars. The kidney damage is correlated with the grade of reflux and the number of UTI, but other factors may also play a role. Uromodulin (UMOD) is a protein produced by kidney tubular cells, forming a matrix in the lumen. We evaluated whether the common variant rs4293393 in the UMOD gene was associated with febrile UTI (FUTI) and/or scars in a group of children with VUR. A total of 31 patients with primary VUR were enrolled. Renal scars were detected in 16 children; no scar was detected in 15 children. Genotype rs4293393 TC (TC) was present in 8 patients, 7 (88%) had scars; genotype rs4293393 TT (TT) was found in 23 patients, and 9 (39%) had scars. Among children with scars, those with TC compared with those with TT were younger (mean age 77 vs. 101 months), their reflux grade was comparable (3.7 vs. 3.9), and the number of FUTI was lower (2.9 vs. 3.7 per patient). Children with VUR carrying UMOD genotype rs4293393 TC seem more prone to developing renal scars, independent of FUTI.

## 1. Introduction

Vesico-ureteral reflux (VUR) is a common congenital anomaly of the urinary tract [1,2]. VUR is associated with urinary tract infections (UTI) [3] which may involve the kidneys by causing a permanent damage known as reflux nephropathy (RN) characterized by renal scars (RS). RN may produce hypertension, complications during pregnancy, and kidney failure [4,5]. The kidney damage correlates with the grade of reflux and the number of UTI, but other factors may have a role and in fact in some cases there is no evidence of UTI A genetic predisposition to VUR and RN is well recognized [1]. Chemoprophylaxis has been widely used to reduce the occurrence of UTI in children with recurrent UTI and or VUR with limited results [1]. The identification of biomarkers that could identify children at risk for recurrent UTI and the understanding of the molecular basis of the increased risk of renal scarring among children with recurrent febrile UTIs are identified as research priorities [6].

Uromodulin (UMOD) is a glycoprotein that is exclusively synthesized in the kidney by the epithelial cells of the thick ascending limb of the loop of Henle (TAL) and the distal convoluted tubule. It is the most abundant protein in healthy human urine. Uromodulin is encoded by the UMOD gene on chromosome 16p12.3.6. The UMOD locus includes single-nucleotide polymorphisms (SNPs) that are in complete linkage disequilibrium in a large block encompassing the gene promoter. Among these SNPs, the rs4293393 UMOD variant has two alleles: T, the major allele associated with risk of hypertension and chronic kidney disease, and C, the minor allele associated with recurrent UTIs [7]. UMOD interacts with components of the immune system [8] and protects against ascending UTI by a binding with type I-fimbriated *Escherichia coli* [9]. A defect of Umod production may increase the susceptibility for UTI [9]. Consequently, studying UMOD genotype in children with FUTI may reveal whether there is a group at increased risk of recurrent UTI and/or RS. The aim of our study was to evaluate whether the distribution of rs4293393 UMOD variants in children is associated with VUR and febrile UTI (FUTI) and/or scars.

## 2. Materials and Methods

### 2.1. Study Patients

Children with VUR followed at our Unit consecutively for 12 months were enrolled. Inclusion criteria were children with documented VUR and RS on DMSA scintigraphy, normal estimated glomerular filtration rate (eGFR) and blood pressure, and no other urinary tract malformation. Patients with secondary VUR, reduced eGFR, high blood pressure were excluded. Age, sex, family history of urinary tract malformations, urinary disorders (leakage of urine, urge-incontinence, urinary urgency, incontinence, enuresis), previous episodes of FUTI, assumption of antibiotic therapy/prophylaxis and surgery for correction of VUR were registered. In all the children age, weight, height, and systolic (SBP) and diastolic (DBP) blood pressure were assessed with validated devices. Blood samples were taken to measure: serum blood urea nitrogen, serum creatinine; DNA extracted for UMOD genotyping. The eGFR was calculated according to the Schwartz formula [10]. Grade of reflux (1 to 5) was established after voiding cystogram. Renal scars were detected on DMSA scan performed at least six months after a UTI by two independent observers. Kidney ultrasonography, cystography and DMSA scintigraphy were carried out, at least six months after UTI, in order to assess the presence and degree of VUR and the presence of RS. Reflux grade when present bilaterally was calculated as the mean of the higher reflux in both ureters.

### 2.2. Determination of UMOD rs42993393 T > C Genotype

Informed consent was obtained from the parents of all our children. Genotyping of SNP for rs4293393 (T/C) was carried out on genomic DNA extracted from white blood cells at LGC Genomics (formerly KBioscience, London, UK) using a competitive allele-specific PCR technique (KASPar v4.0) (call rate 96.4%). Additional quality control criteria included inter- and intra-plate duplicate testing and clear separation of signal clusters [8].

### 2.3. Statistical Analysis

Data were analyzed by the open-source software R: R Core Team (Vienna, Austria, 2021). R: A language and environment for statistical computing (R Foundation for Statistical Computing, Vienna, Austria. URL https://www.R-project.org/ (accessed on 20 December 2022)). The chi square test for independence was used to compare categorical variables. Non-parametric methods (Wilcoxon test for independent groups, Kruskal–Wallis test) were used to compare the distribution of a continuous variable in groups. Regression models were used for further analyses, including multiple logistic regression analysis considering the presence of scars as binary outcome.

## 3. Results

### 3.1. Patient Characteristics

A total of 31 children (age range 7–196, mean 74.7 months) with VUR were enrolled. Reflux was bilateral in 15 and unilateral in 16 children (left kidney, 10 vs. right kidney, 6) respectively. All children except one had reflux grades higher than 2. Kidney scars were detected in 16 children (age range 20–196, mean 89 months): 9 males and 7 females; reflux was grade 3.3, bilateral in 8; FUTI were 3.3 per patient; all except one were on antibiotic prophylaxis and 5 had UTI during treatment; eight children had a surgical correction.

No scar was detected in 15 children (age range 7–169, mean 57.7 months): 8 males and 7 females; reflux grade was 3.5, bilateral in 7; FUTI were 2.0 per patient; one had a surgical correction; all except one were on antibiotic prophylaxis and 3 had UTI during treatment. Table 1 shows the data of the two groups: children with scars were older while gender and reflux grade were comparable; febrile infections were more frequent in patients with scars but the difference did not reach statistical significance. No correlation was found between degree of VUR and percentage of parenchymal loss detectable by the scintigraph study.

### 3.2. Uromodulin Genotyping

Genotype rs4293393 (TC) was present in 8 patients, with 7 (88%) having scars, whereas the genotype rs4293393 (TT), was found in 23 patients, with 9 (39%) having scars. The difference between the occurrence of scars and the genotype at rs4293393 was significant (*p* = 0.018). The reflux grade was comparable in the two groups whereas FUTI were more frequent in the TC group as shown in Table 2 and Table 3.

Among children with scars, those having the TC genotype at rs4293393 with TC compared to those with TT were younger (mean age 77 vs. 101 months), with lower reflux grade (3.7 vs 3.9), and lower number of FUTI (2.9 vs. 3.7 per patient); see Table 3. Multi logistic regression analysis in all patients showed that scars were more common in older children with more FUTI and TC genotype (Table 4)

## 4. Discussion

In this study we evaluated factors which may be related to the formation of scars in children affected by VUR. One of the major problems in the treatment of such patients is to avoid the production of irreversible kidney damage. It has been questioned whether the kidney lesion associated with VUR is congenital or caused by the UTI associated with the reflux [11]. In our cohort of children with VUR almost 50% had kidney scars; this percentage is higher than that reported in other studies, likely because these children were followed in the outpatient clinic which is dedicated to more severe cases. Several risk factors have been described in the production of kidney scars, including reflux grade, number of urinary tract infections, type of bacteria causing the infection, delay in antibiotic treatment, and genetic predisposition. No information on the duration of fever, the type of bacteria involved in each infection, and the treatment was available. Fever associated with UTI is considered a sign of kidney involvement; for this reason we registered only febrile infections (FUTI). We confirm that in our patients the number of FUTI is a predisposing factor to kidney scars, together with older age.

Our study tested the possible association of UMOD genotype and kidney damage in a small group of children with VUR. We found that the UMOD genotype rs423393 TC may be a predisposing factor for kidney scars. Uromodulin is known to show antimicrobial properties in urine; Umod-knockout mice are more prone to UTIs induced by type-1 fimbriated *E. coli* [8]. Furthermore, interstitial uromodulin may cross-talk with proximal tubule cells, contributing to regulate innate immunity, whereas circulating uromodulin may act as an anti-oxidant [12]. Uromodulin regulates sodium transport systems in kidney tubules, thus regulating blood pressure and urinary-concentrating ability. The UMOD locus includes SNP (e.g., rs12917707 and rs4293393) that are in complete linkage disequilibrium in a large block encompassing the gene promoter. Critically, the UMOD promoter variants are associated with the expression of uromodulin in the kidney and its levels in the urine and blood. In fact, homozygous carriers of the UMOD risk T allele (T, major) allele at rs4293393 have twofold higher levels of uromodulin in the urine, compared with homozygous carriers of the (C, minor) protective allele [13]. We limited our UMOD genetic study to rs4293393 which has been found to be related to chronic kidney disease and hypertension; in fact, population genetics investigations indicated that of the top UMOD GWAS variant, rs4293393, associated with CKD risk, is the ancestral allele and is kept at high frequency in most modern populations [14]. The distribution of the UMOD ancestral allele does not follow the ancestral susceptibility model observed for variants associated with salt-sensitive hypertension, i.e., a higher prevalence of salt-retaining alleles in African compared with non-African populations [15]. Instead, the risk variant showed a significant correlation with pathogen diversity (bacteria, helminths) and prevalence of antibiotic-resistant UTIs. An inverse correlation between urinary levels of uromodulin and markers of UTIs has been described [15]. A prospective cohort study of elderly community-dwelling individuals found that those with urinary uromodulin concentrations in the highest quartile had a lower risk of UTI events than those in the lowest quartile, independent of classical UTI risk factors [9].

Here we found that UTIs are more frequent in children carrying TT, but non-significantly. Since TT patients are older, they probably had higher chances to develop a FUTI. No difference was noticed in reflux grade.

Antibiotic prophylaxis has been shown to be effective in reducing the number of UTI but not renal scars and its use has been questioned [16]; whether our observation can be validated will be proved in a larger population in which the prophylaxis could be limited to children with VUR and the rs4293393 TC genotypes.

A few studies investigated the relation between VUR and UMOD. Andriole [17] noticed that interstitial UMOD deposits were frequent in kidneys of patients and animals with focal scarring secondary to VUR. Akioka et al. [18] examined the presence of VUR in 10 transplanted children with interstitial UMOD deposits in kidney allografts; the histological findings of these patients were interstitial mononuclear cell infiltration and fibrosis associated with interstitial UMOD deposits; of note, eight of ten patients (80%) had VUR into the graft and previous UTI. Uto et al. [19] studied the relationship between urinary Uromodulin excretion and eGFR in 26 consecutive patients with primary VUR before and after antireflux surgery; they noticed that low excretion was related to decreased kidney function. To our knowledge no UMOD genetic studies have been performed in patients affected by VUR.

Our study has several limitations. The number of patients was small, although we were able to find a significant difference between patients with scars or without kidney scars. The number of febrile UTI was calculated based on the history referred by parents. Reflux grading may change over the years and our classification may not represent the initial grade. The presence of a scar in a kidney is dependent on the observer, although in our study a radiologist and a nephrologist independently estimated the presence of a scar. Although we excluded children with secondary VUR we did not perform urodynamic studies to detect functional bladder disorders which have also been related to VUR and kidney scars. Finally, we did not correlate genotypes with UMOD levels in the urine of our patients. A prospective study should be done in a larger number of children with VUR.

## 5. Conclusions

Children with VUR carrying the UMOD genotype rs4293393 TC seem more prone to develop kidney scars independently from febrile urinary tract infections.

## Figures and Tables

**Table 1 biomedicines-11-00509-t001:** Clinico-pathological characteristics of the patients.

	SCAR	NO SCAR
N	16	15
M/F	9/7	8/7
Age in months mean (range) *p* = 0.06	90.7 (20–196)	57.7 (7–169)
FUTI (cumulative number) *p* = 0.15	3.3	2
Maximum VUR grade (mean)	3.8	3.5
Bilateral VUR	8/16 (50%)	7/15 (46.7%)

Abbreviations: FUTI, Febrile Urinary Tract Infections; VUR, Vesico-Ureteral Reflux.

**Table 2 biomedicines-11-00509-t002:** Genotype at rs4293393 of Uromodulin in patients with VUR.

Genotype	rs4293393 TC	rs4293393 TT
N	8 (26%)	23 (74%)
M/F	5/3	12/11
Age in months mean (range)	68.3 (4–187)	75.3 (7–169)
FUTI (cumulative number)	2.75	1.78
Maximum VUR grade (mean)	3.6	3.6
Bilateral VUR	3/8 (38%)	12/23 (52%)
No Scar	1	14
Scar *p* = 0.018	7 (88%)	9 (39%)

Abbreviations: FUTI, Febrile Urinary Tract Infections; VUR, Vesico-Ureteral Reflux.

**Table 3 biomedicines-11-00509-t003:** Genotype at rs4293393 of Uromodulin in patients with VUR and scars.

Genotype	rs4293393 TC	rs4293393 TT
N	7	9
Age (months) *p* = 0.61	77	101
Max VUR grade (mean) *p* = 0.82	3.7	3.9
FUTI (number) *p* = 0.55	2.9	3.7

Abbreviations: FUTI, Febrile Urinary Tract Infections; VUR, Vesico-Ureteral Reflux.

**Table 4 biomedicines-11-00509-t004:** Multiple logistic regression analysis considering the presence of scars as outcome.

	*p*
Scars in rs4293393 TC genotype	0.04
Scars correlate with increasing age	0.04
Scars correlate with increasing number of FUTI	0.03
Scars in males	0.19

## Data Availability

Data regarding this study are available on request to the corresponding author.

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
