# Peer review of "Uromodulin and Vesico-Ureteral Reflux: A Genetic Study"

_biomedicines, 2023, doi:10.3390/biomedicines11020509_

Round 1

Reviewer 1 Report

23-01-12-Maringhini 2023 subm biomed-Review-Uromodulin and Vesico-Ureteral Reflux. A Genetic Study

This is a very interesting study in children with VUR. The authors found that children with VUR carrying UMAD genotype rs4293393 TC were more prone to develop renal scars independently from febrile UTI as those with VUR carrying UMOD genotype  rs4293393 TT.

The limitations of the study are well discussed and a prospective study with larger number of children with VUR should be done.

The study is well performed and presented.

Author Response

We thank the reviewer for the very positive feedback. An extensive English revision has been done. 

Reviewer 2 Report

The manuscript entitled “Uromodulin and Vesico-Ureteral Reflux. A Genetic Study” represents an important study of the VUR phenomenon that, although it does not report a particularly large casuistry, is well conducted and supported by sufficient statistical processing to be considered within the available scientific literature on the subject.

The "Introduction" section is perhaps far too concise for how relevant and well-organized it is. I would suggest that the Authors expand it especially by providing more information regarding uromodulin and the UMOD gene.

Materials and methods, results and discussion are conducted with scientific rigor and described in detail by the Authors, although the tables could be more essential in their layout so as to be more readable and usable by the reader.

Although extremely concise, the conclusions report the essence of the research.

Finally, I can conclude that the manuscript is of high quality, and I congratulate the authors on their investigation; I think it simply needs to be enriched in the introductory section regarding the two aspects I mentioned above.

Author Response

We appreciate the reviewer's comments and have modified the text to address all the points. Specifically, more information on Uromodulin has been added in the introduction and the number of tables have been reduced in order to make them easier to consult.